# Recent Developments and Applications of Food-Based Emulsifiers from Plant and Animal Sources

Yuqiao Jin and Achyut Adhikari *

School of Nutrition and Food Sciences, Louisiana State University, Baton Rouge, LA 70803, USA;
yjin@agcenter.lsu.edu
* Correspondence: acadhikari@agcenter.lsu.edu

**Abstract**

Food-based emulsifiers, derived from natural or edible sources such as soybeans, oats, eggs, milk, and fruits, have gained increasing attention in the food industry due to their clean label appeal, recognition as natural ingredients, and alignment with consumer demand for fewer synthetic additives. These emulsifiers are also valued for their biodegradability, environmental sustainability, and potential nutritional benefits. The food-based compounds have been extensively studied for their functional and physicochemical properties. This review provides a comprehensive overview of recent developments and applications of food-based emulsifiers, with a focus on protein-based, polysaccharide-based, and phospholipid-based emulsifying agents derived from plant and animal sources. The mechanisms, advantages, and disadvantages of the food-based emulsifiers are discussed. Plant-based emulsifiers offer sustainability, wide availability, and cost-efficiency, positioning them as a promising area for research. Combinations of food-based emulsifiers such as polysaccharides, proteins, and phospholipids can be utilized to enhance emulsion stability. This paper evaluates current literature and discusses future challenges and trends in the development of food-based emulsifiers.

**Keywords:** emulsion; protein-based emulsifier; polysaccharide-based emulsifier; phospholipid-based emulsifier; food grade



## 1. Introduction

An emulsion is a dispersed system composed of two or more liquid phases that are immiscible with each other [1]. Dispersions are typically classified into two main types: oil-in-water (O/W) and water-in-oil (W/O) systems [1,2]. Emulsions play an important role in the food industry. Examples of oil-in-water emulsions include milk and mayonnaise, while butter and margarine are examples of water-in-oil emulsions. Due to the thermodynamic nature, food emulsions are inherently unstable systems [3]. During transportation and shelf life, food emulsions are prone to destabilization through physical processes such as coalescence, flocculation, and creaming [3]. Emulsifiers, such as proteins and polysaccharides, are surface-active biopolymers capable of stabilizing interfaces and delay the destabilization progress [3,4]. Emulsifiers are generally amphiphilic molecules, containing both polar (hydrophilic) and nonpolar (hydrophobic) regions [3]. This dual-character structure enables them to adsorb at interfaces and form a protective layer around the dispersed phase, enhancing emulsion stability [3,5].

Emulsifiers are widely incorporated into food formulations to enhance stability and texture. Emulsifiers must either be approved as food additives through a formal peti-

tion process or be classified as Generally Recognized as Safe (GRAS) for their intended use. For example, polysorbate 80 (Tween 80) is used as an emulsifier in ice cream to improve the dispersion of fat and prevent fat aggregation, resulting in a smoother texture and better meltdown properties [6]. However, synthetic emulsifiers have been associated with potential health risks and may exhibit toxic effects with prolonged consumption [7]. Some synthetic emulsifiers, such as polysorbates (e.g., Polysorbate 80) and carboxymethyl-cellulose (CMC), have raised concerns due to their potential effects on gut health and metabolism [8,9]. Animal studies have shown that certain synthetic emulsifiers may disrupt the gut microbiota, promote intestinal inflammation, and increase the risk of metabolic syndrome [8,9]. Synthetic emulsifiers have also raised concerns due to their low biodegradability, which contributes to potential toxicity and environmental pollution [7,10,11]. For example, ethoxylated surfactants can release 1,4-dioxane, a probable human carcinogen, during manufacturing or degradation [12]. There is a growing effort to replace synthetic emulsifiers with natural alternatives [5,7].

More consumers seek food products made with natural and sustainable ingredients [13,14]. The global expansion of natural and organic food markets has significantly contributed to the rising demand for food-based emulsifiers. Many retailers are implementing clean-label and free-from standards, encouraging manufacturers to reformulate with food-based alternatives. Proteins, polysaccharides, and phospholipids extracted from foods can be utilized as natural emulsifiers in the formulation of food emulsions [15]. For example, lecithin, a phospholipid commonly used as a commercial food emulsifier, is derived from sources such as soybean oil, eggs, liver, soybeans, peanuts, and wheat germ [16–19]. It can be used in cakes, chocolate, and dairy products as an emulsifier to enhance texture, reduce viscosity, promote an even distribution of ingredients, and prevents destabilization during storage [16–19].

Due to the biodegradability, environmental sustainability, and potential nutritional advantages, food-based emulsifiers have been the focus of extensive research on their functional and physicochemical characteristics. This paper reviews the interfacial adsorption mechanisms, structural characteristics, recent developments, and applications of food-based emulsifiers. Emulsifying agents derived from plant and animal sources, such as proteins, polysaccharides, and phospholipids, are evaluated. The paper also discusses future challenges and emerging trends in the development and use of these emulsifiers.

## 2. Protein-Based Emulsifiers

**Mechanisms**. Proteins are amphiphilic macromolecules composed of one or more polypeptide chains [20]. The ability of proteins to stabilize emulsions and foams arises from their amphiphilic character, which is conferred by the acid-base properties of their constituent amino acids [20,21]. A single protein molecule can possess hydrophilic and hydrophobic domains, both structured and unstructured regions, as well as areas with positive, negative, or neutral charges [21]. Proteins are capable of forming strong viscoelastic films at interfaces, improving both the kinetic stability and thermodynamic of emulsions through hydrogen bonding, hydrophobic interactions, and electrostatic forces [21]. They can form an interfacial coating that inhibits flocculation and coalescence of oil droplets by providing a combination of steric hindrance and electrostatic repulsion (Figure 1A) [21]. Unlike low-molecular-weight emulsifiers that rapidly diffuse to interfaces to facilitate emulsion formation, proteins are macromolecules that diffuse more slowly due to their larger size [22]. Upon reaching the interface, the protein partially unfolds to expose hidden hydrophobic residues, allowing it to reorient with hydrophobic amino acids facing the oil phase and hydrophilic ones facing the aqueous phase [22]. The emulsification capacity, measured as the grams of oil stabilized per gram of protein prior to phase inversion, is gen-

erally lower for proteins compared to small-molecule emulsifiers [22]. Protein emulsifiers serve multiple roles in food products, functioning as fat stabilizers, surfactants, humectants, plasticizers, and crumb softeners in applications such as confectionery, beverages, dairy items, and bakery ingredients [23].

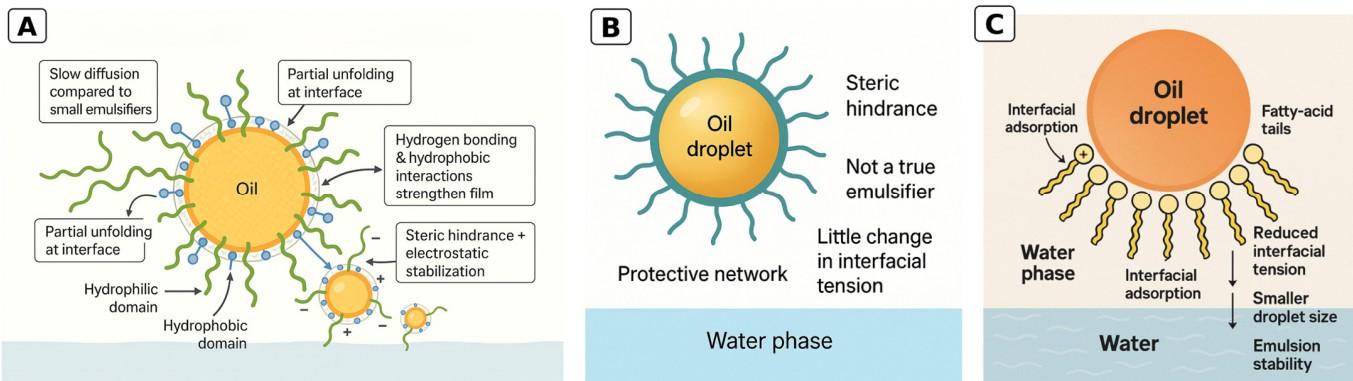

**Figure 1.** Mechanism of (**A**) protein, (**B**) polysaccharide and (**C**) phospholipid-based emulsifiers.

**Plant-derived proteins.** Protein-based emulsifiers have been extensively studied, with plant-derived proteins emerging as an important area of focus. Emulsification is a key functional attribute of plant proteins, as many possess surface-active properties that enable them to function as natural surfactants [22]. Compared to animal proteins, plant proteins typically exhibit a higher proportion of β-sheet structures, fewer α-helices, and a greater presence of fibrillar protein assemblies [24,25]. In plant proteins, most hydrophobic residues are buried within the protein core, and the structure tends to remain relatively rigid after adsorption [22]. This rigidity can limit interactions with adjacent protein molecules, leading to weaker in-plane interactions and decreased stiffness of the interfacial film at the oil–water interface [22]. The interfacial modulus under dilation and shear is generally lower for plant proteins, and their storage modulus exhibits reduced strain dependence, which contributes to the comparatively weaker emulsifying performance of plant proteins versus animal proteins [22,26].

Due to the perceived health benefits and growing consumer acceptance, plant proteins are increasingly used as complete or partial replacements in a variety of food formulations [23]. They enhance the nutritional value of foods by supplying energy and essential amino acids. The increasing demand for clean-label products highlights the need to develop plant-derived proteins as natural emulsifiers [23]. Lists of plant-derived proteins employed as emulsifiers are shown in Table 1. For example, soy proteins emulsifiers are widely used in the food industry. Soy proteins consist of approximately 40% 7S (β-conglycinin) and 30% 11S (glycinin) protein fractions [27]. The 7S globulin exhibits superior emulsifying properties compared to the 11S globulin, primarily because the 11S globulin exists in a more stable oligomeric form [27]. Soy protein can stabilize oil-in-water emulsions in various food products, such as sausages, yogurt, ice cream, and coffee whiteners [27]. Soy proteins are regarded as promising base materials for developing functional emulsifiers with tailored physicochemical properties through various enzymatic, biological, chemical, and physical modification techniques [27,28].

**Animal-derived proteins.** Animal-derived proteins are widely used as emulsifiers in the food industry due to their amphiphilic nature, which enables them to adsorb at oil–water interfaces, lower interfacial tension, and stabilize oil droplets by preventing aggregation [29]. The animal protein, such as whey, primarily consisted of small-sized proteins, while the plant protein ingredients, such as soy protein, contained a comparatively higher proportion of larger proteins [30]. Smaller animal proteins diffuse quickly to the oil–

water interface and form stable films, enhancing emulsion stability [22]. In contrast, larger plant proteins have limited mobility and interfacial activity due to their rigid, aggregated structure [30,31]. Animal-derived proteins may show better heat, salt, and pH stability, compared with plant-derived proteins [30]. For example, Tan et al. (2022) reported that both whey protein and soy protein coated oil droplets exhibited aggregation near their isoelectric points and under high ionic strength conditions [30]. Without added salt, whey protein droplets aggregated extensively between pH 4 and 5, and soy protein droplets from pH 2 to 5 [30]. The isoelectric points for both were around pH 5 [30]. The whey protein-coated droplets showed a higher magnitude of the surface potential compared to soy protein coated samples, indicating a larger number of anionic groups per unit surface area [30]. At pH 7, significant aggregation occurred in soy protein-coated droplets at NaCl concentrations of 100 mM or higher, whereas whey protein-coated droplets only aggregated at concentrations of 400 mM or above [30]. Lists of animal-derived proteins employed as emulsifiers are shown in Table 1. For example, milk is a naturally occurring oil-in-water emulsion in which fat droplets are dispersed throughout the aqueous phase [23]. Dairy proteins (casein and whey protein) have been utilized as natural emulsifiers in the production of ice cream, cheese, and butter [23].

Insect-based proteins are emerging as sustainable and functional alternatives to traditional protein emulsifiers in food applications [32–35]. Proteins extracted from insects such as crickets, mealworms, and black soldier fly larvae have shown good emulsifying activity and emulsion stability in oil-in-water systems [32–35]. These proteins can adsorb at the oil–water interface and reduce interfacial tension, similar to conventional emulsifiers like soy or whey protein [32–35]. Additionally, insect proteins are rich in essential amino acids and offer environmental benefits such as low land and water usage and reduced greenhouse gas emissions, aligning with clean-label and sustainable food trends [32–35]. Trujillo-Cayado et al. (2024) reported that cricket protein combined with rhamsan gum can form stable emulgels with small droplet sizes, making them promising for use in encapsulation system [36].

**Modification methods**. With advances in technology, proteins can be modified to improve the functional properties. Physical modifications such as heat treatment, high-pressure processing, and ultrasound can disrupt native protein structure and unfold protein to expose hydrophobic groups [37–39]. Li et al. (2023) treated wheat germ protein with high-intensity ultrasound (20 kHz) at varying power levels ranging from 200 to 800 W [39]. Ultrasound processing led to the unfolding of the protein's molecular structure [39]. This was evidenced by an increase in surface hydrophobicity and surface free sulfhydryl group levels, along with a decrease in intrinsic fluorescence intensity [39]. Results showed that ultrasound treatment enhanced the absorption of wheat germ protein at the oil–water interface and decreased the interfacial tension [39]. Treatment of wheat germ protein at 400 W for 20 min is recommended to achieve optimal results [39]. Chemical modification involves covalent alterations to protein molecules. For instance, acylation and succinylation introduce charged groups that improve solubility and molecular flexibility [40]. Cross-linking agents such as glutaraldehyde and enzymes like transglutaminase enhance emulsion stability by forming larger protein networks [41,42]. Enzymatic treatments, including proteolysis and transglutaminase modification, can also improve solubility and interfacial activity [43]. Kim et al. (2025) studied the interfacial behavior of pea-whey protein after enzymatic cross-linking with microbial transglutaminase and the addition of maltodextrin [44]. At the oil–water interface, enzymatic cross-linking reduced the viscosity and elasticity, accompanies by gradual and continuous adsorption [44]. The combination of enzymatic cross-linking and maltodextrin addition resulted in superior emulsion stability,

effectively preventing flocculation and coalescence [44]. Biological or genetic modifications enable the production of proteins with tailored emulsifying properties [45].

In practice, combinations of different modification methods are often employed to achieve optimal emulsifying performance. For example, as the low solubility limits the application of wheat gluten protein in food products, Xiong et al. (2023) modified the wheat gluten protein by pH cycling with heat treatment (80 °C) [46]. The solubility of wheat gluten protein increased from 6.7% to 71.1% after treatment [46]. Higher solubility improves the protein's ability to disperse uniformly in aqueous environments, enhancing its availability at the oil–water interface [46–48]. This facilitates faster adsorption, formation of more cohesive interfacial films, and better stabilization of emulsions [46–48]. As the wheat gluten modified by pH-shifting treatment combined with heat showed better solubility and emulsibility, it can be applied to ice cream [46,47]. Ice cream formulated with modified wheat gluten showed favorable sensory attributes, including desirable color and firmness [47].

**Advantages and disadvantages.** Protein-based emulsifiers offer several advantages over synthetic ones, including enhanced stability, irreversible interfacial adsorption, the ability to form thick interfacial layers, improved nutritional value, being natural and sustainable, and well-suited for clean-label food products [20]. Proteins can be modified or combined with other ingredients to enhance their emulsifying performance. However, proteins are sensitive to low pH and high temperatures, which can compromise their stability and functionality. Protein-coated oil droplets tend to be unstable at pH levels near their isoelectric point and under conditions of high ionic strength [49]. The food industry is increasingly reformulating its products by replacing functional ingredients that are chemically synthesized or obtained from animal sources such as eggs, fish, meat, fish, or milk with ingredients derived from plants [29]. This shift is largely driven by the growing demand for foods that support animal welfare, human health, and environmental sustainability [29]. The plant protein-based emulsifier is a growing trend due to their environmental sustainability and potential health benefits [22]. One of the challenges is finding suitable alternative plant-derived proteins that can match the functional properties of animal proteins in specific food applications. For plant-based proteins, the development of large aggregates and phase separation can occur if environmental conditions are not properly regulated [30]. Some plant-based proteins are associated with allergenic reaction, which pose health risks. There is also lack of standardized analytical methods for evaluating and comparing the functional properties of animal and alternative proteins under conditions relevant to actual food applications [29].

**Table 1.** Examples of protein-based emulsifiers investigated in recent research.

| Emulsifier | Food Source | Emulsion Type | Emulsion Properties | Oil Phase and Applications | Ref. |
|---|---|---|---|---|---|
| RuBisCo protein (ribulose 1,5-bisphosphate carboxylase) | Duckweed (*Lemna minor*) | O/W | Dispersed phase volume fraction: 10% ($w/w$) Emulsifier concentration: 1% ($w/w$) Dispersed phase droplet size: 0.91 μm | Oil phase: soybean oil. Application: beverage, dressing, sauce, dip | [30] |
| Soy protein | Soybean | O/W | Dispersed phase volume fraction: 10% ($w/w$) Emulsifier concentration: 1% ($w/w$) Dispersed phase droplet size: 0.37 μm | Oil phase: soybean oil. Application: beverage, dressing, sauce, dip | [30] |
| Soy protein | Soybean | W/O/W | Dispersed phase volume fraction: 20% ($w/w$) Emulsifier concentration: 1% ($w/w$) Dispersed phase droplet size: 22–25 μm | Oil phase: soybean oil. Application: double emulsion | [50] |
| Succinylated soy protein | Soybean | O/W | Dispersed phase volume fraction: 10% ($mL/mL$) | Oil phase: corn oil | [51] |
| Rice protein | Rice | O/W | Dispersed phase volume fraction: 3% ($w/w$) Emulsifier concentration: 0.6% ($w/w$) Dispersed phase droplet size: 1.71 μm (pH2), 1.79 μm (pH 10) | Oil phase: soybean oil | [52] |

**Table 1.** *Cont.*

| Emulsifier | Food Source | Emulsion Type | Emulsion Properties | Oil Phase and Applications | Ref. |
|---|---|---|---|---|---|
| Rice protein | Rice | O/W | Dispersed phase volume fraction: 10% (*w/w*) Emulsifier concentration: 1% (*w/w*) Dispersed phase droplet size: 19.48 μm | Oil phase: linseed oil | [53] |
| Rice protein hydrolysates | Rice | O/W | Dispersed phase volume fraction: 10% (*w/w*) Emulsifier concentration: 1% (*w/w*) Dispersed phase droplet size: 10.15–16.69 μm | Oil phase: linseed oil | [53] |
| Rice protein fibril | Rice | O/W | Emulsifier concentration: 0.09% (*w/v*) | Oil phase: corn oil | [54] |
| Pea protein | Pea | O/W | Dispersed phase volume fraction: 10% (*w/w*) Emulsifier concentration: 0.5% (*w/w*) | Oil phase: rapeseed oil | [55] |
| Wheat germ protein (ultrasound treated) | Wheat | O/W | Dispersed phase volume fraction: 20% (*v/v*) Dispersed phase droplet size: 0.29 μm | Oil phase: soybean oil | [39] |
| *Zanthoxylum* seed protein (ultrasound) | *Zanthoxylum* | O/W | Dispersed phase volume fraction: 25% (*v/v*) Dispersed phase droplet size: 0.7896 μm | Oil phase: soybean oil | [56] |
| Whey protein | Milk | O/W | Dispersed phase volume fraction: 10% (*w/w*) Emulsifier concentration: 1% (*w/w*) Dispersed phase droplet size: 0.25 μm | Oil phase: soybean oil. Application: beverage, dressing, sauce, dip | [30] |
| Whey protein | Milk | W/O/W | Dispersed phase volume fraction: 20% (*w/w*) Emulsifier concentration: 1% (*w/w*) Dispersed phase droplet size: 22–25 μm | Oil phase: soybean oil. Application: double emulsion | [50] |
| Whey protein | Milk | W/O | Dispersed phase volume fraction: 15.17% (*w/w*) | Oil phase: milk fat. Application: butter | [57] |
| Whey protein fibril | Milk | O/W | Dispersed phase volume fraction: 20% (*v/v*) Emulsifier concentration: 2% (*w/v*) Dispersed phase droplet size: 10–100 μm | Oil phase: soybean oil | [58] |
| Whole milk powder | Milk | W/O | Dispersed phase volume fraction: 15.19% (*w/w*) | Oil phase: milk fat. Application: butter | [57] |
| Soy protein isolate and young apple polyphenol | Soybean and apple | O/W | Dispersed phase volume fraction: 20% (*v/v*) | Oil phase: rapeseed oil. Application: nano-deliver functional oils and nutrients | [59] |
| Soy protein isolate and tea polyphenol conjugates | Soybean and tea | O/W | Dispersed phase volume fraction: 25% (*v/v*) Emulsifier concentration: 0.3% (*w/v*) Dispersed phase droplet size: 496 nm | Oil phase: soybean oil | [60] |
| Casein butyrylated dextrin complex nanoparticle | Milk | O/W | Dispersed phase volume fraction: 30% (*v/v*) | Oil phase: corn oil | [61] |
| Sodium caseinate/phloretin complexes | Milk | O/W | Dispersed phase volume fraction: 20% (*v/v*) | Oil phase: sunflower oil | [62] |
| Sodium caseinate and maltodextrin | Milk | O/W | Dispersed phase volume fraction: 3% (*v/v*) Emulsifier concentration: 4% (*w/v*) Dispersed phase droplet size: 201.7–602.7 nm | Oil phase: peanut oil body | [63] |
| Cricket protein and rhamsan gum | Cricket | O/W | Dispersed phase volume fraction: 10% (*w/w*) Emulsifier concentration: 2% (*w/w*) Dispersed phase droplet size: 0.688 μm | Oil phase: avocado oil | [36] |

## 3. Polysaccharide-Based Emulsifiers

**Mechanisms**. In contrast to protein-based particles that reduce interfacial tension and form a viscoelastic film at the interface, polysaccharides stabilize emulsions primarily by forming a protective network around droplets in O/W emulsion, offering steric hindrance to prevent coalescence [64–67]. Polysaccharides do not function as true emulsifiers in the formation of O/W emulsions [66,68]. Rather, they serve as stabilizers by forming a thick, gel-like steric barrier around oil droplets, thereby preventing droplet coalescence (Figure 1B) [66,68]. Polysaccharides are typically biocompatible, biodegradable, non-toxic, readily available, and exhibit stability across a broad range of temperatures and pH levels [66].

**Plant-derived polysaccharides.** Most polysaccharide-based emulsifiers are derived from plant sources due to low cost, easy scalability, and fewer allergen and dietary concerns. Plant polysaccharides are natural biopolymers abundantly present in plant cell structures [69]. Natural plant-based polysaccharides usually have limited emulsifying

activity since they lack hydrophobic–hydrophilic balance and do not absorb efficiently at the oil–water interface [64]. They tend to stabilize emulsions by increasing viscosity or forming steric barriers, instead of reducing interfacial tension [64]. Examples of plant-based polysaccharide used as emulsifiers are listed in Table 2. For example, the mucilage of okra is a highly viscous polysaccharide composed of highly hydrophilic, high molecular weight substances [70]. Noorlaila et al. (2015) investigated the emulsifying properties of okra mucilage and its potential application as an emulsifier in coconut milk [71]. Okra mucilage was prepared using a water extraction method, and its viscosity was evaluated at various temperatures (10, 30, 50, and 70 °C), along with assessments of its water-holding capacity, oil-holding capacity, emulsifying capacity, and emulsion stability [70]. When added into the coconut emulsion, it resulted in a yield of mucilage (1.46%) and an oil-holding capacity of 854.25 g/100 g [71]. The advantages of okra mucilage include its biocompatibility, non-toxicity, low cost, and abundant natural availability [71]. It was reported that Okra mucilage has the potential to serve as a natural emulsifier, particularly in food emulsion systems [71].

Recent technological advancements have led to the development of various green extraction techniques, such as pulsed electric fields, ultrasound, microwave-assisted extraction, ohmic heating, high-pressure processing, subcritical water extraction, and enzyme-assisted methods [72]. These methods share common advantages such as increasing the yield of polysaccharide, reducing acid usage, and shortening reaction time [72]. By minimizing environmental impact and improving efficiency, they support the sustainable production of high-quality emulsifiers suitable for food applications. For example, among pectin extraction methods, conventional heating combined with acid catalysts to solubilize protopectin is one of the most widely used approaches [72]. Costa et al. (2025) employed microwave-assisted extraction to isolate pectin from apple pomace [73]. Chan et al. (2023) utilized ultrasound-assisted extraction of pectin from jackfruit [74]. Results underscored both the practical and economic benefits of using modern technologies such as microwave heating and ultrasound for pectin extraction [73,74].

**Animal (insect)-derived polysaccharides.** Most animal-derived polysaccharides are not widely used in food emulsification because of the limited availability, high production cost, and regulatory restrictions. However, chitosan derived from the exoskeletons of insects such as crickets and mealworms have gained attention as natural emulsifiers in food systems [75,76]. Chitosan is a cationic polysaccharide with excellent film-forming, emulsifying, and antimicrobial properties [75,76]. As a biodegradable and renewable biopolymer, insect-sourced chitosan offers a sustainable alternative to marine-derived chitosan and aligns with clean-label food trends [75,76]. Casariego et al. (2024) studied chitosan as a natural emulsifier in mayonnaise and reported that it exhibited a notable foaming capacity [75].

**Modification methods**. Modification (physical, chemical, or biological) can be utilized to improve interfacial activity, introduce hydrophobic groups, and improve interaction with protein or lipids [75]. The modification of polysaccharides using chemical, physical, biological or enzymatic methods to enhance their emulsifying performance and explore their structure–activity relationships have been widely developed [67]. Structural modifications of natural emulsifiers can significantly influence the interfacial behavior and bulk properties [77]. Chemical modification involves the covalent attachment of functional groups to alter a polysaccharide's solubility and amphiphilic balance [67]. Hydrophobization, such as with octenyl succinic anhydride, is widely applied to starch and gum arabic to introduce hydrophobic moieties that enhance surface activity [78]. Carboxymethylation and sulfation introduce negatively charged groups, improving solubility and providing electrostatic stabilization in emulsions [79]. Wei et al. (2020) utilized corn fiber gum, a polysaccharide-

based emulsifier modified with varying levels of octenyl succinic anhydride, to achieve a wide range of interfacial properties [77]. It was reported that esterified corn fiber gum, in comparison to native corn fiber gum, formed thicker interfacial films with increased elasticity and viscosity, leading to enhanced stability of the resulting emulsions [77]. Huang et al. (2024) modified chitosan with phenolic acids and observed improvements in sustained release rate, encapsulation efficiency, and photostability [80]. Physical modification alters the structural properties of polysaccharides without changing their chemical bonds. For example, heat or shear treatment can modify molecular conformation or reduce molecular weight, thereby affecting viscosity and interfacial behavior [81]. Techniques such as ultrasound and high-pressure homogenization further break down high-molecular-weight polysaccharides to enhance their emulsifying performance [82]. Enzymatic modifications, including depolymerization and transglycosylation, can selectively cleave or restructure polysaccharide chains to improve functionality [83]. Xu et al. (2025) studied the interfacial properties of soy hull polysaccharide at the oil–water interface and the enzyme-mediated stabilization mechanism of high internal phase emulsions [84]. The polysaccharide's secondary structure underwent conformational rearrangement, with partial unfolding upon enzyme addition to the emulsion system, suggesting that soy hull polysaccharide adsorbs at the oil–water interface to form a robust interfacial membrane that enhances emulsion stability [84]. Biological modification enables the production of novel emulsifiers with enhanced interfacial activity and tailored structural properties.

**Advantages and disadvantages.** Polysaccharides are highly versatile and can be readily tailored or modified to meet the requirements of specific applications [67,85,86]. Polysaccharide-based emulsifiers are low cost, easy scalability, and fewer allergen and dietary concerns. It meets the demand for label-friendly food-grade emulsifiers [87]. Plant polysaccharides are abundant in plant cell structures [69]. While widely used as natural stabilizers in emulsions, polysaccharides have limited effectiveness as true emulsifiers. Its surface activity is low because most native polysaccharides are highly hydrophilic, which limits the ability to reduce interfacial tension. Many polysaccharides need chemical, physical, or biological modification to enhance the wettability [64]. Combining the properties of polysaccharides and proteins under optimal conditions, such as appropriate protein-to-polysaccharide ratio, ionic strength, concentration, pH, and temperature, has proven to be an effective strategy for improving emulsion formation and stability [7,88].

**Table 2.** Examples of polysaccharide-based emulsifiers investigated in recent research.

| Emulsifier | Food Source | Emulsion Type | Emulsion Properties | Oil Phase and Applications | Ref. |
|---|---|---|---|---|---|
| Mucilage | Okra | O/W | Dispersed phase volume fraction: 9.1% ($v/v$) | Oil phase: corn oil. Application: coconut milk | [71] |
| Pectin | Apple pomace | O/W | Emulsifier concentration: 4% ($w/w$) | Application: food gel | [73] |
| Pectin | Jackfruit | O/W | Emulsifier concentration: 2% ($w/v$) | Application: food gel | [74] |
| Gum arabic | Acacia Senegal tree | W/O/W | Emulsifier concentration: 1% ($w/w$) Dispersed phase droplet size: 22–25 μm | Oil phase: soybean oil. Application: double emulsion | [50] |
| Corn fiber gum | Corn | O/W | Dispersed phase volume fraction: 5% ($w/w$) Emulsifier concentration: 0.5–1.5% ($w/w$) Dispersed phase droplet size: >2.5 μm | Oil phase: soybean oil | [77] |
| Corn fiber gum (modified with octenyl succinic anhydride) | Corn | O/W | Dispersed phase volume fraction: 5% ($w/w$) Emulsifier concentration: 0.5–1.5% ($w/w$) Dispersed phase droplet size: 1.11–2.50 μm | Oil phase: soybean oil | [77] |

**Table 2.** *Cont.*

| Emulsifier | Food Source | Emulsion Type | Emulsion Properties | Oil Phase and Applications | Ref. |
|---|---|---|---|---|---|
| Rice flour | Rice | O/W | Emulsifier concentration: 2.45% ($w/w$) | Application: cooked sausage | [89] |
| Tapioca starch | Tapioca | O/W | Emulsifier concentration: 2.45% ($w/w$) | Application: cooked sausage | [89] |
| Seaweed powder | Seaweed | O/W | Emulsifier concentration: 1% ($w/w$) | Application: dairy products | [90] |
| Chitosan | Exoskeleton of insects | O/W | Emulsifier concentration: 0.1–0.5% ($w/w$) | Application: mayonnaise | [75] |
| Quillaja saponin | Quillaja | O/W | Dispersed phase volume fraction: 10% ($w/w$) Emulsifier concentration: 0.5-2.5% ($w/w$) Dispersed phase droplet size: 0.15–0.5 μm | Oil phase: medium chain triglyceride oil. Application: non-dairy creamer | [91] |
| Potato starch and nanoliposomes | Potato | O/W | Dispersed phase volume fraction: 20% ($v/v$) Dispersed phase droplet size: 7–20 μm | Oil phase: soybean oil | [92] |
| Tapioca starch and milk protein | Cassava and milk | O/W | Emulsifier concentration: milk protein 10.5%, Modified tapioca starch 0–2% | Oil phase: anhydrous milk fat | [93] |
| Protein-polysaccharide conjugates | Sugar beet pulp | O/W | Emulsifier concentration: 1% ($w/w$) Dispersed phase droplet size: 0.438–0.479 μm | Oil phase: medium-chain triglycerides | [87] |

## 4. Phospholipid-Based Emulsifiers

**Mechanisms**. Phospholipids have a distinctive molecular structure comprising a lipophilic region formed by fatty acid chains and a hydrophilic region consisting of phosphate-based ester groups [94]. Phospholipids tend to align at the oil–water interface, where they reduce interfacial tension and decrease the pressure gradients needed to break up droplets during the emulsification process [95]. The amphiphilic structure can significantly lower the interfacial tension between oil and water, thereby promoting the formation of a uniform and stable emulsion (Figure 1C) [96,97]. Owing to their amphiphilic nature, phospholipids serve as effective emulsifiers by reducing droplet size in oil-in-water emulsions, contributing to improved emulsion stability [96]. Phospholipid-based emulsifiers can be found naturally within foods such as soybeans, peanuts, rapeseeds, sunflower kernel, wheat germ, eggs, liver, and milk [16,17]. Plant-derived phospholipids primarily consist of phosphatidylcholine, phosphatidylethanolamine, and phosphatidylinositol [96]. Due to their high content of unsaturated fatty acids, they exhibit lower thermal stability and are more prone to oxidation [98]. In contrast, animal-derived phospholipids mainly include phosphatidylcholine, phosphatidylethanolamine, sphingomyelin, and lysophospholipids [99]. Their higher proportion of saturated fatty acids contributes to greater thermal stability and improved resistance to oxidation [100].

**Plant-derived phospholipids.** Plant-derived phospholipids are widely used for their affordability, functional performance, and broad consumer acceptance. Examples of Phospholipid-based natural emulsifiers are shown in Table 3. Züge et al. (2017) explored the use of phospholipids extracted from avocado pulp oil as natural emulsifiers [101]. Emulsions were prepared with varying oil contents (20–70 vol%) and phospholipid concentrations (1–2 wt%) at pH 3 and 7 [101]. Results showed that phospholipid concentration had a greater impact on emulsion stability and droplet size than pH [101]. Emulsions with 2 wt% phospholipids exhibited gel-like, pseudoplastic behavior, and those with 50–60 vol% oil were most stable [101]. Rinaldi et al. (2014) evaluated the influence of three types of phospholipid emulsifiers (soy, rice, and milk phospholipids) on the thermal, physicochemical, and sensory properties of a basic Italian gelato formulation, with mono- and diglycerides of fatty acids used as a reference emulsifier for comparison [102]. It was reported that gelato formulated with soy phospholipids showed increased firmness compared to other samples [102]. In food applications, combinations of phospholipids with other emulsifiers have been explored to improve emulsion stability [94,103]. Wang et al. (2025) investigated

the molecular mechanism by which taurine enhances the interaction between soy lecithin and rice starch [104]. It was reported that the addition of taurine improved the thermal stability of the complex, resulting in a 16% increase in resistant starch and a 14% increase in slowly digestible starch [104].

**Animal-derived phospholipids.** Animal-derived phospholipids, particularly those extracted from egg yolk and milk, are widely used as natural emulsifiers in food. For example, egg yolk lecithin helps slow the aging process, protects the stomach and liver, enhances the absorption of fat-soluble vitamins, and promotes more efficient blood circulation [105,106]. Egg yolk lecithin serves as a fundamental emulsifier in a wide range of food products, including mayonnaise, salad dressings, ice cream, cheese spreads, chocolate, and various baked goods. Phospholipids can be combined with other emulsifiers to enhance emulsion stability [94,103]. For example, the formation of protein–phospholipid complexes can enhance the coverage of the oil droplet surface and increase the thickness of the interfacial layer [107]. García-Moreno et al. (2014) investigated the effects of combining casein (0.3% w/w) and phospholipids (0.5% w/w) as emulsifiers on the physical and oxidative stability of 10% fish oil-in-water emulsions (pH 7) [107]. It was reported that the combination of casein and lecithin contributed to a favorable interfacial layer structure and thickness, which effectively inhibited lipid oxidation in the emulsion [107].

Insect-based phospholipids are gaining attention as novel natural emulsifiers for food applications due to their functional similarity to traditional sources like egg and soy lecithin [108–110]. Extracted primarily from insect fat bodies or tissues such as those of black soldier fly larvae and mealworms, these phospholipids exhibit strong surface activity and can effectively stabilize oil-in-water emulsions [108–110]. Li et al. (2024) extracted lecithin from black soldier fly larvae and yellow mealworm, identifying insects as a novel source of lecithin for food applications [110].

**Modification methods**. Enzymatic approaches are widely used to modify the fatty acid composition or polar head groups of phospholipids, improving functionality such as emulsifying capacity, oxidative stability, or thermal resistance [111,112]. For example, phospholipase A2 or phospholipase D can hydrolyze or transfer groups in the phospholipid molecule [113,114]. Xu et al. (2024) reported that after phospholipase A2 treatments, the solubility (83.49%) of egg yolk was significantly higher compared to that of untreated egg yolk (27.89%) [114]. Chemical methods can introduce functional groups or alter the fatty acid chains to tailor emulsification behavior [115]. Physical treatments can affect phospholipid packing, dispersibility, or crystallization behavior. For example, high-pressure homogenization or sonication can change the particle size of lecithin dispersions, improving emulsion droplet size and stability [116]. Advanced methods involve using genetically engineered microbes or controlled fermentation to produce custom phospholipids with defined structures.

**Advantages and disadvantages.** Phospholipids are abundant, naturally derived, and biocompatible, making them attractive emulsion stabilizers [97]. Phospholipid-based food emulsifiers are considered as clean-label ingredients. The use of phospholipid mixtures with co-surfactants or proteins to stabilize emulsions has attracted considerable interest [11,103]. The cost of phospholipid-based food emulsifiers can be more expensive than synthetic emulsifiers. Some ingredients can raise allergen issues. Phospholipids are highly unstable when exposed to air or sunlight, beginning to discolor at around 80 °C and undergoing oxidation at temperatures of 100 °C or higher, eventually leading to decomposition [96]. Lipid oxidation, the development of off-flavors, and reduced shelf life are concerns [117]. The use of co-emulsifiers with antioxidant properties may enhance oxidative stability [117].

**Table 3.** Examples of phospholipid-based emulsifiers investigated in recent research.

| Emulsifier | Food Source | Emulsion Type | Emulsion Properties | Oil Phase and Applications | Ref. |
|---|---|---|---|---|---|
| Soy phospholipids | Soybean | O/W | Emulsifier concentration: 0.35% (*w/w*) | Application: gelato | [102] |
| Soy phospholipids | Soybean | O/W | Dispersed phase volume fraction: 10% (*v/v*) Dispersed phase droplet size: 381.27 nm | Application: infant formula | [118] |
| Milk phospholipids | Milk | O/W | Emulsifier concentration: 0.35% (*w/w*) | Application: gelato | [102] |
| Milk phospholipids | Milk | O/W | Dispersed phase volume fraction: 10% (*v/v*) Dispersed phase droplet size: 334.5 nm | Application: infant formula | [118] |
| Rice phospholipids | Rice | O/W | Emulsifier concentration: 0.35% (*w/w*) | Application: gelato | [102] |
| Pulp oil | Avocado | O/W | Dispersed phase volume fraction: 20–70% (*v/v*) Emulsifier concentration: 1, 2% (*w/w*) Dispersed phase droplet size: 9.47–64.49 μm | Oil phase: soybean oil | [101] |
| Soy lecithin | Soybean | O/W | Dispersed phase volume fraction: 9.65% (*w/w*) Emulsifier concentration: 0.5-2.0% (*w/w*) | Oil phase: rice bran oil | [119] |
| Sunflower phospholipids | Sunflower seed | O/W | Dispersed phase volume fraction: 10% (*v/v*) Dispersed phase droplet size: 378.97 nm | Application: infant formula | [118] |
| Sesame lecithin | Sesame oil | W/O | Dispersed phase volume fraction: 27% water (*v/v*) | Application: margarine | [120] |
| Corn lecithin | Corn oil | W/O | Dispersed phase volume fraction: 3% water (*v/v*) | Oil phase: corn oil. Aqueous phase: water. | [121] |
| Egg yolk phospholipids | Egg | O/W | Dispersed phase volume fraction: 10% (*v/v*) | Application: infant formula | [118] |
| Lecithin | Yellow mealworm | O/W | Extraction yield: 4–10% | Application: general food | [110] |
| Lecithin | Black soldier fly larvae | O/W | Extraction yield: 4–10% | Application: general food | [110] |
| Soy lecithin, rice starch, and taurine | Soybean and rice | O/W | Emulsifier concentration: 40 mg/mL soy lecithin in ethanol, added to 4% rice starch emulsion | Application: general food | [104] |
| Casein and lecithin | Milk and soybean | O/W | Dispersed phase volume fraction: 10% (*w/w*) Emulsifier concentration: 0.3% casein and 0.5% lecithin (*w/w*) Dispersed phase droplet size: 210 μm | Application: fish oil | [107] |

## 5. Hybrid Emulsifiers

Comparison of protein, polysaccharide, and phospholipid emulsifiers are shown in Table 4. Hybrid emulsifier systems, particularly those combining proteins, phospholipids, and polysaccharides, have been widely studied because of their synergistic effects on emulsification, stability, and functionality in complex food systems. For example, polysaccharides can be used in combination with other emulsifiers, such as in polysaccharide–protein mixed systems [67]. Proteins exhibit strong adsorption at the oil–water interface when at their isoelectric point, thereby contributing to emulsion stabilization [122,123]. However, proteins may aggregate at the oil–water interface when the pH is near their isoelectric point or under conditions of high ionic strength [123]. Adding polysaccharides to protein aggregates can stabilize oil-in-water emulsions [122]. There are three main types of polysaccharide–protein complexes [124]. The first type is naturally occurring complexes, where protein residues are covalently bound to polysaccharide chains [124]. The second type is Maillard conjugates, which are formed through a covalent bond between the reducing end of a polysaccharide and an amine group on a protein [124]. The third type is electrostatic complexes, which result from interactions between polysaccharides and proteins carrying opposite net charges [124]. The use of polysaccharide–protein complexes to stabilize oil-in-water emulsions has attracted growing interest in recent years [7,124,125]. For example, Lin et al. (2023) obtained alkaline-extracted protein–polysaccharide conjugates from partially depectinized sugar beet pulp using hot alkaline extraction [87]. The macromolecule exhibits a structural arrangement with a protein core surrounded by a polysaccharide shell [87]. Emulsion droplet sizes were 0.438 μm for the fresh emulsion and 0.479 μm after storage

at 60 °C for 5 days [87]. It significantly reduced the oil–water interfacial tension (11.58 mN/m) and demonstrated excellent emulsifying stability (10%) [87]. It was reported that alkaline-extracted protein–polysaccharide conjugates effectively stabilized gel-like high internal phase emulsions (oil fraction 0.80) at a low concentration of 0.2%, even under high ionic strength conditions ranging from 0 to 1000 mM [87].

**Table 4.** Comparison of protein, polysaccharide and phospholipid emulsifiers.

| Class | Interfacial Behavior | Stability Tendencies | Pros and Cons | Modification Methods and Examples |
|---|---|---|---|---|
| Proteins | Adsorb and partially unfold at the interface, forming a viscoelastic protein film. Provide steric hindrance and electrostatic repulsion. Produce a moderate decrease in interfacial tension compared with small-molecule surfactants. | Generally stable at moderate pH and ionic strength. Least stable near the isoelectric point. Salts can screen charges. Thermal history can denature or strengthen films. | Pros: strong interfacial films; clean label; can achieve small droplets with suitable processing. Cons: sensitive near the isoelectric point; ionic effects; slower diffusion than small surfactants; potential allergens depending on source. | Physical: heat, pH cycling, ultrasound, high pressure. Chemical: acylation or succinylation, polyphenol conjugation, glycation. Enzymatic: transglutaminase crosslinking or limited hydrolysis. |
| Polysaccharides | Do not act as classic emulsifiers. Form thick gel-like shells or networks around droplets that provide steric stabilization and reduce coalescence. Increase continuous-phase viscosity. Little direct effect on interfacial tension. | Often robust to pH and temperature (polymer dependent). Charge, substitution, molecular weight, and branching control barrier thickness and sensitivity to salts. | Pros: biocompatible and biodegradable; label friendly; improve creaming stability and mouthfeel. Cons: often require a co-emulsifier for droplet formation; higher viscosity can affect processing and sensory properties. | Physical: shear, heat, pH. Chemical: octenyl succinic anhydride modification, carboxymethylation or acetylation, protein complexation or coacervation. Enzymatic: controlled depolymerization or crosslinking. |
| Phospholipids | Amphiphilic small molecules with a polar head and hydrophobic tails that align at the interface and markedly lower interfacial tension. Facilitate droplet breakup during homogenization and produce fine droplets. | Stability is strongly influenced by fatty-acid saturation. Unsaturated phospholipids are more prone to oxidation and thermal degradation, whereas saturated fractions are more thermally and oxidatively stable. | Pros: effective at low use levels; produce fine droplets; broadly compatible. Cons: oxidation and thermal sensitivity for unsaturated fractions; possible flavor interactions; variability by source and fraction. | Blending of headgroups (PC, PE, PI, SM, lyso-PL), use of antioxidants, fractionation. Enzymatic hydrolysis to lyso-PL can increase solubility and hydrophilic–lipophilic balance. |

Phospholipids help enhance the dispersibility and uniformity of emulsions [126]. Researchers demonstrated that interactions between phospholipids and proteins significantly affect emulsification performance [125,127]. For example, phospholipids can interact with the hydrophobic amino acids of whey protein isolate, altering its secondary structure [126]. An increase in the β-sheet content disrupts intra-chain hydrogen bonds, while additional hydrogen bonding between peptide chains is associated with protein unfolding [126,128]. It was reported that combing phospholipids with whey protein led to a decrease in particle size (from 232.33 nm to 188.59 nm) and a significant improvement in emulsion stability (from −34.26 mV to −49.13 mV) [126]. When combined with proteins, phospholipids, or polysaccharides, these biopolymers form multi-layered interfacial structures and mixed networks in the continuous phase, leading to better emulsification efficiency and long-term stability than single emulsifiers alone [126,129,130].

## 6. Conclusions

Emulsifiers play an essential role in the food industry. Growing consumer demand for environmentally friendly products made from natural and sustainable ingredients has driven increased interest in using natural emulsifiers as alternatives to synthetic ones.

Proteins, polysaccharides, and phospholipids are among the most commonly used natural emulsifiers in the formulation of food emulsions. Comparative analysis of the physical and chemical properties of food-based and synthetic emulsifiers, along with the development of food-based emulsifier applications in food systems, is needed. Creating customized food emulsifiers tailored to specific brands can be of significant importance. Plant-derived emulsifiers are sustainable, abundant, and cost-effective, making them a promising focus for future research. Developing plant-based emulsified food products represents an emerging trend in the field. The sustainable production of insects, requiring low land and water resources, aligns with growing consumer demand for eco-friendly and functional food ingredients.

No single emulsifier can fully satisfy the diverse requirements because each application demands specific functional properties. These requirements include emulsion stability over time and under varying pH, temperature, or ionic conditions; desired texture and mouthfeel; sensory attributes such as taste and appearance; compatibility with other ingredients; and compliance with clean-label or nutritional expectations. Different emulsifiers excel in different areas. Combining different food-based emulsifiers may lead to improved emulsification performance.

Emerging sources of food-based proteins, including microbial and algal origins, should be thoroughly investigated, with particular focus on novel extraction and cultivation technologies to improve yield. Future studies are needed to design optimized blends of emulsifiers that can deliver enhanced texture, stability, rheological behavior, and other critical functional properties. Research should also emphasize structural modifications of food-based emulsifiers to strengthen their emulsification performance. Since different modification approaches yield distinct outcomes, comparative studies are necessary. While lab-scale research has explored ingredients isolated from various food sources, future studies are necessary at pilot and industrial scales to facilitate process optimization and scale-up. At the industrial level, production strategies should be developed to overcome inherent challenges such as undesirable color and flavor, ensuring both functionality and consumer acceptance.

**Author Contributions:** Writing—original draft preparation, and review and editing, Y.J. and A.A.; supervision and project administration, A.A.; funding acquisition, A.A. All authors have read and agreed to the published version of the manuscript.

**Funding:** This work was supported by the USDA NIFA HATCH project #LAB94565.

**Data Availability Statement:** No new data were created or analyzed in this study. Data sharing is not applicable to this article.

**Conflicts of Interest:** The authors declare no conflicts of interest.

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
