# Peer review of "Recent Developments and Applications of Food-Based Emulsifiers from Plant and Animal Sources"

_colloids, doi:10.3390/colloids9050061_

Round 1
Reviewer 1 Report (Previous Reviewer 3)
Comments and Suggestions for Authors
Thank you for the modifications. I will recommend the manuscript for publication.
Author Response
Thank you for all your help to make this manuscript better.
Reviewer 2 Report (New Reviewer)
Comments and Suggestions for Authors
Please refer to the attachment for the reviewer's recommendation.

Author Response
- Suggestion: It is suggested to includes figure for mechanism of each emulsifiers. Mechanisms are described but sometimes say at a descriptive level rather than deeply linking molecular structure to interfacial properties. Suggestion: Include more structure-function diagrams showing how different molecular arrangements (e.g., β-sheet vs α-helix content, degree of glycosylation) influence interfacial adsorption and stability. Figure on m mechanism of stabilization for each emulsifier type/ modification pathways (physical, chemical, enzymatic).
Reply: Figure 1 included to illustrate the mechanism of polysaccharide, protein and phospholipid emulsifiers
- Can the authors expand on hybrid emulsifier systems (e.g., protein-phospholipid-polysaccharide) and their synergistic effects.
Reply: Thanks for the comment. Hybrid emulsifier systems were added in Section 5.
- These tables have the same structure and mainly differ by the type of emulsifier used (protein, polysaccharide, or phospholipid). They give useful examples of recent studies, sources, and applications, but the format makes it hard to compare the three types directly. Suggestion: Add one more table comparing the main mechanism of these emulsifiers side by side, including how they stabilize emulsions, typical stability ranges, advantages/limitations, and common modification methods. This would help readers quickly see the functional differences between the three groups.
- Modification methods. The explanation of modification methods is brief and does not clearly describe how each method improves emulsifier performance.
The section also cites a limited number of references, and more recent studies could be added to provide a fuller picture of current advancements and their functional implications.
Reply: Thank you very much for the suggestion. More explanations and references are added.
- The manuscript explains each emulsifier type proteins, polysaccharides, and phospholipids, in separate sections, but there is no comparative analysis across these categories. Without side-by-side comparison, it is difficult for readers to identify which emulsifier performs best under specific parameters such as droplet size range, stability at varying pH/ionic strength, thermal tolerance, or cost-effectiveness. Including a comparative discussion or summary table highlighting these differences would add significant value and help guide emulsifier selection for different food applications.
Include how emulsifiers can be carriers for bioactives, impact nutrient bioavailability, and contribute to functional foods.
Reply: Table 4 added as suggested.
- In the conclusion, the authors list key requirements for emulsifiers, including stability over time, tolerance to pH, temperature, and ionic changes, as well as texture, sensory attributes, ingredient compatibility, and clean-label compliance. However, these parameters are not systematically addressed in the discussion section, nor is there a table summarizing them for different emulsifier types. The discussion only briefly touches on some of these aspects, without providing comparative data or structured analysis. A dedicated table or matrix comparing how protein-, polysaccharide-, and phospholipid-based emulsifiers perform against these parameters would strength the manuscript and align the discussion more closely with the stated conclusion.
Reply: The requested information added in the new Table 4.
- ensure consistent abbreviation formatting (e.g., O/W, W/O/W) and units (µm vs micron)
Reply: Done
- the conclusion is descriptive but does not clearly highlight where research should go next.
Reply: Thanks for the comment. “Future research” is added in the conclusion.
This manuscript is a resubmission of an earlier submission. The following is a list of the peer review reports and author responses from that submission.
Round 1
Reviewer 1 Report
Comments and Suggestions for Authors
1. The language is not concise enough, with repetitive expressions of the same ideas, such as the amphiphilicity of proteins, membrane formation at interfaces, and the demand for plant proteins, among others. These are not the only instances.
2. The logic is also unclear. For example, the source and structural characteristics of polysaccharides (Line 258) and phospholipids (Lines 318-323) should be placed at the beginning of the section. These are not the only issues.
3. Modification methods for proteins, polysaccharides, etc., including physical and chemical approaches, are merely listed as examples. Instead, the principles, patterns, or conclusions of typical modification methods should be described.
4. The authors categorize the content by raw materials but should specifically outline the characteristics, modification methods, and effects of typical plant proteins, animal proteins, polysaccharides, etc.
Author Response
- The language is not concise enough, with repetitive expressions of the same ideas, such as the amphiphilicity of proteins, membrane formation at interfaces, and the demand for plant proteins, among others. These are not the only instances.
Reply: Thanks for the comment. The manuscript has been thoroughly revised.
- The logic is also unclear. For example, the source and structural characteristics of polysaccharides (Line 258) and phospholipids (Lines 318-323) should be placed at the beginning of the section. These are not the only issues.
Reply: Thanks for the comment. Information on the source and structure of polysaccharides and phospholipids has been moved to the beginning of each section.
- Modification methods for proteins, polysaccharides, etc., including physical and chemical approaches, are merely listed as examples. Instead, the principles, patterns, or conclusions of typical modification methods should be described.
Reply: Thanks for the comment. Additional details have been incorporated into the section on modification methods for proteins, polysaccharides, and other relevant substances.
- The authors categorize the content by raw materials but should specifically outline the characteristics, modification methods, and effects of typical plant proteins, animal proteins, polysaccharides, etc.
Reply: Thanks for the comment. The outlines have been updated and clarified in the manuscript.
Reviewer 2 Report
Comments and Suggestions for Authors
Comments 1: Lines 15–17: I suggest the authors to remove or simplify the repeated mention of “protein-based, polysaccharide-based, and phospholipid-based emulsifying agents,” as this has already been clearly stated in Line 14. Merging these sentences would improve conciseness and avoid unnecessary redundancy in the abstract.
Comments 2: Lines 46–48: I suggest the authors to further clarify or support the statements regarding the potential health risks and environmental concerns associated with synthetic emulsifiers. Providing specific examples or citing more recent studies would enhance the credibility of these claims and help contextualize the need for natural alternatives.
Comments 3: Lines 127–129: Consider briefly explaining the practical significance of this solubility increase in terms of emulsifier performance.
Comments 4: Lines 134–140: Consider elaborating briefly on the functional implications of differences in protein size and structure for emulsification performance.
Comments 5: Lines 213–219: Consider briefly summarizing the common purpose or advantages of these green extraction techniques to improve paragraph coherence.
Comments 6: Lines 315–317: Consider briefly framing this as a broader application to avoid disrupting the emulsification focus, as the paragraph shifts abruptly from emulsion stability to starch digestibility.
Comments 7: Lines 343–344: Consider briefly specifying what those “diverse requirements” are (e.g., stability, texture, sensory attributes).
Author Response
Comments 1: Lines 15–17: I suggest the authors to remove or simplify the repeated mention of “protein-based, polysaccharide-based, and phospholipid-based emulsifying agents,” as this has already been clearly stated in Line 14. Merging these sentences would improve conciseness and avoid unnecessary redundancy in the abstract.
Reply: Thanks for the comment. The sentences have been revised accordingly in Line 15-17.
Comments 2: Lines 46–48: I suggest the authors to further clarify or support the statements regarding the potential health risks and environmental concerns associated with synthetic emulsifiers. Providing specific examples or citing more recent studies would enhance the credibility of these claims and help contextualize the need for natural alternatives.
Reply: Thanks for the comment. Specific examples have been updated in the manuscript in Line 46-53.
Comments 3: Lines 127–129: Consider briefly explaining the practical significance of this solubility increase in terms of emulsifier performance.
Reply: Thanks for the comment. The practical significance of the increased solubility has been added in Line 186-189.
Comments 4: Lines 134–140: Consider elaborating briefly on the functional implications of differences in protein size and structure for emulsification performance.
Reply: Thanks for the comment. The differences in protein size and structure related to emulsification performance have been added in Line 127-130.
Comments 5: Lines 213–219: Consider briefly summarizing the common purpose or advantages of these green extraction techniques to improve paragraph coherence.
Reply: Thanks for the comment. A discussion of the common purposes and benefits of the green extraction techniques has been incorporated in Line 248-251.
Comments 6: Lines 315–317: Consider briefly framing this as a broader application to avoid disrupting the emulsification focus, as the paragraph shifts abruptly from emulsion stability to starch digestibility.
Reply: Thanks for the comment. The sentences and paragraphs have been reorganized for improved clarity and flow.
Comments 7: Lines 343–344: Consider briefly specifying what those “diverse requirements” are (e.g., stability, texture, sensory attributes).
Reply: Thanks for the comment. Information has been added to briefly specify what the “diverse requirements” refer to in Line 429-432.
Reviewer 3 Report
Comments and Suggestions for Authors
The review entitled ‘‘Recent Development and Application of Food-Based Emulsifiers’’ presents the food emulsifiers currently used in research from animal and plant proteins to polysaccharides to phospholipids. I suggest major revisions before ethe review is reconsidered.
- I suggest the title be reviewed since only animal and plant sources emulsifiers are explored.
- Line 65: What mechanisms?
- Line 141: “protein coated” what?
- Line 145: what is higher surface potential?
- Lines 164-165: These words have been repeated several times already
- Lines 170-185: Why go back to talking about plant proteins after having moved on to animal proteins?
- Lines 176-178: you make it seem as if dairy protein is not suitable for Pickering emulsions’ stabilization.
- Tables in general: Since we are talking about emulsifiers and their emulsifying power as a function of their source, I would have liked to see more physical properties in the tables (like dispersed phase droplet size all through the table as well as emulsifier concentration used and dispersed phase volume fraction). In the column labeled “properties”, it is sometimes confusing to know if we’re talking about the properties of the emulsion or those of the emulsifier.
- Table 1: there are no recent articles working with sodium caseinate?
- Line 198: remove the verb “are”.
- Line 209: yield of what?
- Line 216: yield of what? Protein extraction?
- Line 341: “… are sustainable, abundant, and cost-effective”. Is this the case for long considering global warming and loss of agricultural land? Aren’t there other alternatives that you haven’t explored like insect proteins?
Author Response
The review entitled ‘‘Recent Development and Application of Food-Based Emulsifiers’’ presents the food emulsifiers currently used in research from animal and plant proteins to polysaccharides to phospholipids. I suggest major revisions before ethe review is reconsidered.
I suggest the title be reviewed since only animal and plant sources emulsifiers are explored.
Reply: Thanks for the comment. The title has been updated, and the manuscript revised accordingly.
Line 65: What mechanisms?
Reply: This paper reviews the interfacial adsorption mechanisms, structural characteristics, recent developments, and applications of food-based emulsifiers. The information has been updated in Line 69.
Line 141: “protein coated” what?
Reply: Thanks for the comment. Protein coated oil droplets. The information has been updated in Line 132.
Line 145: what is higher surface potential?
Reply: Thanks for the comment. The sentence has been revised in Line 135.
Lines 164-165: These words have been repeated several times already
Reply: Thanks for the comment. The sentence has been revised.
Lines 170-185: Why go back to talking about plant proteins after having moved on to animal proteins?
Reply: Thanks for the comment. The paragraph has been restructured, and outlines have been added.
Lines 176-178: you make it seem as if dairy protein is not suitable for Pickering emulsions’ stabilization.
Reply: Thanks for the comment. The sentence and paragraph has been restructured.
Tables in general: Since we are talking about emulsifiers and their emulsifying power as a function of their source, I would have liked to see more physical properties in the tables (like dispersed phase droplet size all through the table as well as emulsifier concentration used and dispersed phase volume fraction). In the column labeled “properties”, it is sometimes confusing to know if we’re talking about the properties of the emulsion or those of the emulsifier.
Reply: Thanks for the comment. Tables have been revised to include more physical properties.
Table 1: there are no recent articles working with sodium caseinate?
Reply: Thanks for the comment. Recent research on sodium caseinate has been added to Table 1.
Line 198: remove the verb “are”.
Reply: Removed.
Line 209: yield of what?
Reply: Thanks for the comment. yield of mucilage. The sentence has been revised in Line 240.
Line 216: yield of what? Protein extraction?
Reply: Thanks for the comment. increasing the yield of polysaccharide. The sentence has been revised in Line 248.
Line 341: “… are sustainable, abundant, and cost-effective”. Is this the case for long considering global warming and loss of agricultural land? Aren’t there other alternatives that you haven’t explored like insect proteins?
Reply: Thanks for the comment. Information on insects has been added in the manuscript.